# INVERSE DECISION MAKING VIA INVERSE GENERATIVE MODELING

## ABSTRACT

It is often extremely challenging to infer novel concepts in decision making, such as new actions, goals, or plans, from just a few examples. In this work, we formulate the problem of inferring unfamiliar concepts in decision making as Inverse Decision Making via Inverse Generative Modeling (IDM-IGM). We then introduce a novel concept inference method for this new formulation, which can swiftly adapt to new decision making concepts by leveraging invertible neural generative models. The core idea is to pretrain a generative model on a set of basic concepts and their demonstrations. During test time, given a few demonstrations of a new decision making concept (such as a new goal or a new action), our method can conveniently infer the underlying concept through backpropagation thanks to the invertibility of the generative model. This critically avoids any fine-tuning, greatly accelerating the speed of concept learning. We evaluate our method in three domains – object rearrangement, goal-reaching, and motion caption of human actions. Our experimental results demonstrate that the pretrained generative model can successfully (1) infer learned concepts and generate agent motion or plans of inferred concepts in unseen environments and (2) infer new compositions of learned concepts or even novel concepts to interpret unseen agent behaviors.

## 1 INTRODUCTION

Autonomous agents' behaviors are governed by their beliefs, goals, and internal models of the world. The ability to infer such intent (which we refer to as *concepts*) behind the behaviors of other agents and subsequently generalize them to new unseen situations is a crucial building block for intelligent agents. This ability allows an agent to i) infer other agents' decision making processes and ii) learn to perform new tasks from other agents (including humans). Concept inference in decision making is especially complex, however, as it requires much more than just memorizing patterns in state and action sequences – accurate inference requires an understanding of the internal goals of the agent as well as a model of the causal dynamics of the environment. To successfully infer a concept of an agent moving towards an area in a room, an agent must both infer the desired goal position as well as a model of motion in the environment, so that it can synthesize motions following that concept when the agent enters both unseen states as well as environments with additional constraints on motion.

Concept inference in decision making is a highly underspecified problem, causing existing work to make unrealistic assumptions to enable effective progress. Existing works in inverse reinforcement learning (IRL) (Ho & Ermon, 2016; Fu et al., 2018) typically make the assumption that only provided demonstrations have positive reward in an environment, with all behavior that is not demonstrated providing no reward at all. In a different vein, other work has relied on pretraining on task families of concepts and assumes that concept inference simply corresponds to inferring similar concepts to ones already seen in the task family (Duan et al., 2017; Finn et al., 2017).

To solve this complex challenge, we propose Inverse Decision Making via Inverse Generative Modeling (IDM-IGM). In our approach, we first pretrain a large conditional generative model which synthesizes different decision making trajectories conditioned on different task descriptions. To infer new concepts in decision making from a limited number of demonstrations, we then formulate concept inference in decision making as an *inverse generative modeling problem*, where we find the latent task description in the generative model which maximizes the likelihood of the demonstrations. This approach allows us to leverage the powerful task priors learned by a generative model to infer the shared concepts between demonstrations. We show that such an inferred concept rep-

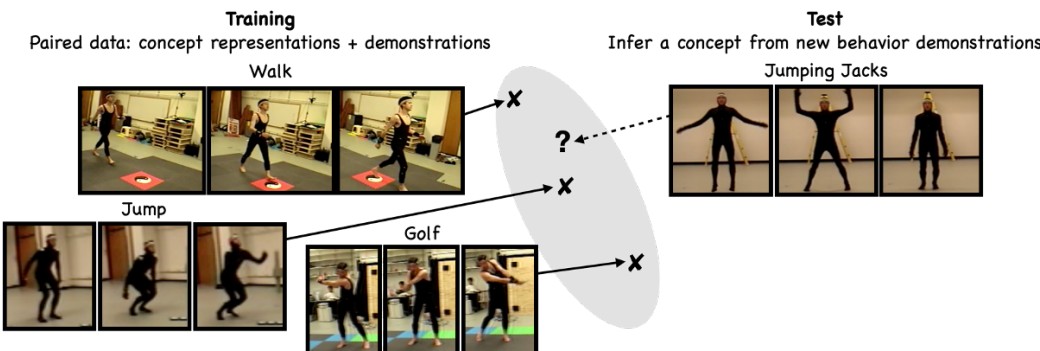

Figure 1: **Concept Inference from Demonstrations.** Given paired data of demonstrations of a task (*e.g.*, walking, jumping and playing golf) and a concept (a latent representation of the task) we learn to generate behavior from concepts. At inference time, given demonstrations of a new concept (*e.g.*, jumping jacks), our aim is to infer the concept representing that task and then use it to generate similar behavior in different settings.

resentation is highly robust and generalizable, and is able to capture both behaviors that have been seen during pretraining (*e.g.*, 'walk', 'jump', 'golf' in Figure 1) but can also *generalize* to completely unseen behavior (*e.g.*, 'jumping jacks'). In contrast, existing approaches towards learning from demonstrations are largely limited to behavior that has been previously seen.

In addition to being able to directly infer a single concept from demonstrations, IDM-IGM can also infer compositions of concepts from demonstrations, *i.e*, infer concepts that are either compositions of seen training concepts (*e.g.*, multiple desired relations between objects) or novel concepts (*e.g.*, new human motion) for various decision making domains: object rearrangement, goal-reaching, and motion capturing. Such inferred concepts are robust and can be composed with other inferred concepts at test time to synthesize novel unseen behavior. Compared to baselines, we show that our approach can generate a more diverse range of trajectories encapsulating the learned concept. We achieve this due to two properties of generative models. First, these models have shown strong interpolation abilities which allow generating the new concept on new initial states they were not demonstrated from (Ramesh et al., 2021; Saharia et al., 2022). Second, these models have compositional properties that enable generating the learned concept composed with other concepts (Liu et al., 2022; Ajay et al., 2023).

Our main contributions are (1) formulating concept inference in decision making as Inverse Decision Making via Inverse Generative Modeling (IDM-IGM), (2) proposing a novel method for efficient decision making concept inference based on the new formulation, and (3) a systematic evaluation revealing the ability of our method to compose and generalize across a diverse set of domains.

## 2   RELATED WORK

**Learning from demonstrations.** Our problem setting is closely related to learning from demonstrations. There has been much work on learning to generate agent behavior given demonstrations of said behavior. There are several common approaches to this problem. First, behavior cloning (BC) is a supervised learning method to learn a policy from demonstrations. Similar to our framework, BC can be goal-conditioned and applied to state-only demonstrations (Ding et al., 2019). BC often suffers from covariate shift (Shimodaira, 2000) and fails to generate the demonstrated behavior in novel scenarios. Second, the inverse reinforcement learning (IRL) framework learns a policy that maximizes the return of an explicitly (Ng et al., 2000; Fu et al., 2018) or implicitly (Ho & Ermon, 2016) learned reward function. These works learn a reward for a single task or for a set of tasks (*e.g.*, goal-conditioned IRL (Bahdanau et al., 2019; Fu et al., 2019) and multi-task IRL (Gleave & Habryka, 2018)). While IRL is more data efficient than BC, it is computationally costly due to learning a policy every iteration via an inner reinforcement learning (RL) loop. Additionally, when faced with a new task, we have to retrain the reward again. A third approach is inverse planning (Baker et al., 2009; 2017; Zhi-Xuan et al., 2020), which can robustly infer concepts such as goals and beliefs even in unseen scenarios. However, it assumes access to a planner and knowledge about environment dynamics, and the task/goal space.

In contrast, we do not learn an action-generating policy directly or via a reward function. We also do not assume having access to any given planner, world model, or prior over the task/goal space. Instead, we learn *concepts* from demonstrations via a pretrained generative model that takes the concept as input and directly produces state sequences. We can then input the learned concept to the generative model to produce similar yet diverse behavior to the demonstrated one. The idea of concept learning via generative models has been explored for computer vision applications (Gal et al., 2023; Liu et al., 2023). We build on this work and show how to extend it to decision making domains. Our work also differs from prior works on learning trajectory representations (Ozair et al., 2021; Nasiriany et al., 2019; Co-Reyes et al., 2018; Hausman et al., 2018; Garg et al., 2022). These works focus on learning plans over trajectory embeddings, whereas we learn a task representation from demonstrations on which we condition to generate behavior.

**Generative Models in Decision Making.** There has been work on generative modeling for decision-making, including generative models for single-agent behaviors, such as implicit BC (Florence et al., 2022), Decision Diffuser (Ajay et al., 2023), Decision Transformer (Chen et al., 2021; Jiang et al., 2023b; Sudhakaran & Risi, 2023), and for multi-agent motion prediction such as Jiang et al. (2023a). There have also been energy-based models for learning reward functions, such as energy-based imitation learning (EBIL) (Liu et al., 2021). In this work, we utilize a conditional generative model for the *inverse* problem, *i.e.*, inferring concepts from demonstrations.

## 3 FORMULATION

Consider an environment with state space $\mathcal{S}$ and action space $\mathcal{A}$, and transition function $\mathcal{T} : \mathcal{S} \times \mathcal{A} \rightarrow \mathcal{S}$. Given a small demonstration dataset $D$ of state-action pairs $\tau = \{(s_0, a_0), (s_1, a_1), ...\}$, we are interested in inferring a behavioral policy $\pi(a|s)$ which generalizes the behavioral capture in $D$. There exist two main approaches to infer behavioral policy $\pi(a|s)$ to match the behavior in the demonstration dataset.

**Imitation Learning (IL).** In imitation learning, we infer policy $\hat{\pi}(a|s) : \mathcal{S} \times \mathcal{A} \rightarrow [0, 1]$ under a supervised Maximum Likelihood Estimation (MLE) objective $\mathbb{E}_{s,a \sim D}[\log \pi(a|s)]$.

**Inverse Reinforcement Learning (IRL).** Unlike IL, IRL learns a reward function $\hat{r}(s_i)$ from the demonstrations $D$ and the corresponding optimal policy $\hat{\pi}$. There have also been recent approaches, such as adversarial maximum-entropy IRL, that directly recover the policy. This objective can be formulated as minimizing the $f$-divergence between the marginal state-action distributions of the estimated and expert policies $D_f(\rho^{\hat{\pi}}(s, a) || \rho^{\pi_{\exp}}(s, a))$ (Ghasemipour et al., 2020).

Inspired by recent success in large generative models, we propose an alternative generative formulation for inferring specific behavioral policy $\pi(a|s)$ given a small set of demonstrations $D$. We term this, **Inverse Decision Making via Inverse Generative Modeling (IDM-IGM)**. In our formulation, we first assume access to a large pretraining dataset $D_{\text{pretrain}}$ of *state*-based sequences $\tau = \{s_0, s_1, ...\} \subseteq T$. We assume that each trajectory $\tau_i$ in $D_{\text{pretrain}}$ is partially annotated with meta-data "concept" $c_i$ describing the trajectory, $\mathcal{D}_{\text{pretrain}} = \{(\tau_i, c_i)\}_{i=1}^N$, where $c_i \in \mathcal{C} \subseteq \mathbb{R}^n$, shapes our understanding of trajectories $\tau_i$. Given $D_{\text{pretrain}}$, we learn a conditional generative model $\mathcal{G}_\theta : \mathcal{C} \times \mathcal{S} \rightarrow T$ conditioned on concepts and initial states, which learns to generate future trajectories. We train the parameters of $\mathcal{G}_\theta$ to maximize likelihood $\arg\max_\theta \mathbb{E}_{\tau, c \sim D_{\text{pretrain}}}[\log \mathcal{G}_\theta(\tau|c, s_0)]$.

At test time, given a demonstration dataset $D$, we formulate inferring a policy $\pi(a|s)$ that can be used to sample trajectories from $D$ as *inverting* the generative model. In particular, we infer a new concept $\tilde{c}$ so that our conditional generative model $\mathcal{G}$ maximizes the likelihood of trajectories in $D$, corresponding to $\arg\max_{\tilde{c}} \mathbb{E}_{\tau, c \sim D_{\text{pretrain}}}[\log \mathcal{G}_\theta(\tau|\tilde{c}, \tau_0)]$. We find this design choice enables us to leverage the priors learned by $\mathcal{G}$ from $D_{\text{pretrain}}$ to effectively infer policies from $D$ given very few demonstrations, even if the demonstrated $D$ deviates very strongly from the concept labels $c$ seen in $D_{\text{pretrain}}$.

The key difference between our approach to infer behavioral policies and prior approaches in behavior cloning and IRL is the assumption and usage of large pretraining dataset of behaviors $D_{\text{pretrain}}$. By learning a generative model across such a dataset, we learn strong priors about the nature of behaviors in these domains, enabling us to more effectively infer new behavior given only a very limited number of demonstrations of the relevant concept. As new inferred behaviors do not have to be $D_{\text{pretrain}}$, such an assumption is often not prohibitive in practice. There is typically a vast amount

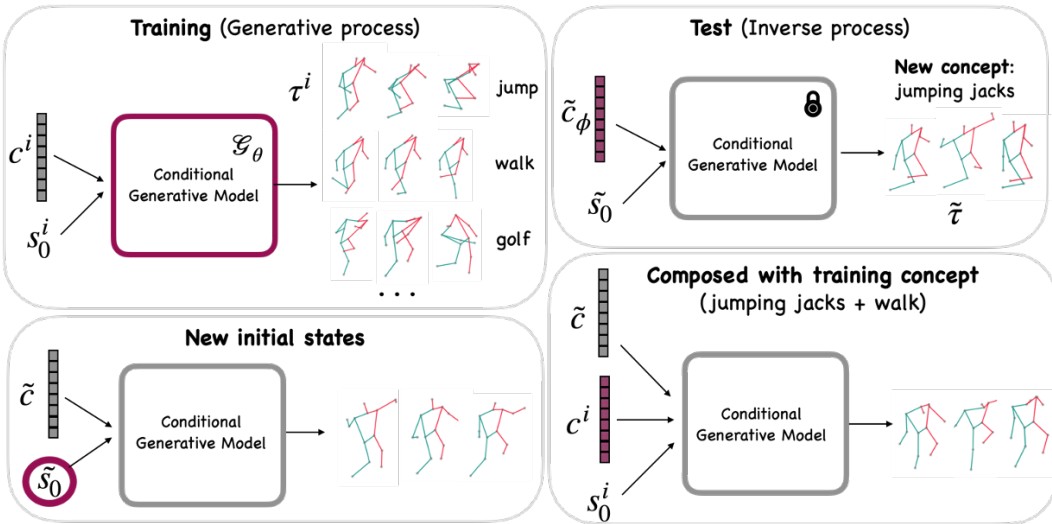

Figure 2: **Concept Inference from Demonstrations via Generative Modeling.** We train a conditional generative model on a set of paired behaviors (*e.g.*, walking, jumping and golfing) and their concept representations. At test time, given few demonstrations of a new behavior (*e.g.*, jumping jacks), we infer a concept that generates the test behavior. We then generate versions of the test behavior conditioned on the learned concept and (1) new initial states (2) composed with other concepts.

of existing trajectories collected from the internet or prior exploration in an environment, which only need to be weakly annotated to characterize the trajectory, *e.g.*, the goal state.

## 4 CONCEPT INFERENCE BASED ON IDM-IGM

We develop a new concept inference method based on the IDM-IGM framework, as outlined in Figure 2. During training we learn a generative model $\mathcal{G}_\theta$ from training $\{(\tau_i, c_i)\}_i$ pairs. At test time we freeze $\mathcal{G}_\theta$, and given demonstrations of a new task $\{\tilde{\tau}\}_i$, optimize a concept $\tilde{c}_\phi$ to produce the test behavior via $\mathcal{G}_\theta$. We then generate a diverse set of behaviors via $\mathcal{G}_\theta$, either for the inferred concept $\tilde{c}_\phi$ conditioned on new initial states or for the compositions of $\tilde{c}_\phi$ with other concepts.

### 4.1 TRAINING A DIFFUSION MODEL TO GENERATE BEHAVIOR

Diffusion models have recently shown success as generative models for decision making domains (Janner et al., 2022; Ajay et al., 2023). Specifically, Ajay et al. (2023) used a conditional classifier-free guidance diffusion model (Ho & Salimans, 2022) to generate trajectories. We adopt their objective and learn a denoising model $\epsilon_\theta$ conditioned on latent concepts and initial states to estimate noise:

$$\mathbb{E}_{\epsilon\sim\mathcal{N}(0,I),k\sim\mathcal{U}(K),(\tau,c)\sim\mathcal{D},\beta\sim\mathrm{Bern}(p)}[||\epsilon - \epsilon_\theta(x_k(\tau),(1-\beta)c+\beta c_\emptyset, s_0, k)||^2] \quad (1)$$

where noise $\epsilon$ is sampled from a normal distribution, $k$ is a timestep sampled uniformly from the number of diffusion timesteps $K$, $p$ is the probability of removing conditioning information which is then replaced by dummy condition $c_\emptyset$, and $s_0$ is the initial state corresponding to $\tau$. $x_k(\tau)$ is obtained from $x_0 = \tau$ by the forward noising process at diffusion step $k$, $q(x_k|x_0) := \mathcal{N}(x_k; \sqrt{\alpha_k}x_0, (1-\alpha_k)I)$ for variance schedule $\alpha_k \in \mathbb{R}$ and by sampled noise $\epsilon$.

### 4.2 INFERRING A NEW CONCEPT FROM DEMONSTRATIONS

Given a trained diffusion model $\epsilon_\theta$ and demonstrations of a new concept $\{\tilde{\tau}\}_i$, we infer a concept $\tilde{c}_\phi$ that best describes the demonstrations. Starting from a uniformly sampled concept embedding $\tilde{c}_\phi \sim \mathcal{U}([0,1]^n)$, we freeze $\epsilon_\theta$, and optimize $\phi$ based on Eq. 1 where $c = \tilde{c}_\phi$ and $\tau \sim \{\tilde{\tau}\}_i$.

A new concept may be described better as a composition of simpler concepts that lie within the training distribution. Instead of inferring one concept $\tilde{c}_\phi$, it is also possible to learn several concepts $\{\tilde{c}_\phi^1, \tilde{c}_\phi^2, ...\}$ and their weights $\{\omega_\psi^1, \omega_\psi^2, ...\}$, and condition on their weighted composition. In this

case the estimated noise in Eq. 1 is replaced by

$$\hat{\epsilon} = \epsilon_\theta(x_k(\tau), c_\emptyset, s_0, k) + \sum_i \omega_\psi^i(\epsilon_\theta(x_k(\tau), \tilde{c}_\phi^i, s_0, k) - \epsilon_\theta(x_k(\tau), c_\emptyset, s_0, k)). \quad (2)$$

### 4.3 GENERATING THE INFERRED CONCEPT

The reverse noising process is calculated based on $\epsilon_\theta$ (Ho et al., 2020). After learning a concept $\tilde{c}_\phi$, we evaluate the behavior it generates by initializing $x_K(\tau) \sim \mathcal{N}(0, \alpha I)$, and compute $x_k \sim \mathcal{N}(\mu_{k-1}, \alpha\Sigma_{k-1})$ iteratively as a function of the estimated noise $\hat{\epsilon}(\epsilon_\theta)$, where $\mu$ and $\Sigma$ are the mean and variance that define the reverse process, until generating $x_0 = \tau$. The noise is estimated by fixed or learned weights as defined in Eq. 2, and by any number of concepts $\geq 1$. The conditition for the generation can be summarized as:

**Inferred concept and demonstrated initial states.** This condition may lead to diverse behaviors in domains where the initial state is irrelevant for a task due to the randomness in sampling $x_K(\tau)$. However, in domains where the initial state and concept jointly determine the optimal behavior, the generation would not be diverse.

**Inferred concept and novel initial states.** We compute $\hat{\epsilon}$ based on novel test states as well to generate more diverse behavior. Imitation learning methods may suffer from covariate shift in this case (Ho & Ermon, 2016). We empirically show that our method is less prone to this problem.

**Inferred concept composed with other concepts.** We add another term to the sum in Eq. 2 where the inferred concept is composed with a training concept $c^i$ and its weight $\omega^i$: $\omega^i(\epsilon_\theta(x_k(\tau), c^i, s_0, k) - \epsilon_\theta(x_k(\tau), c_\emptyset, s_0, k))$, corresponding to the composition of factors (Liu et al., 2022).

## 5 EXPERIMENTS

We demonstrate results in three domains: object rearrangement goal-oriented navigation, and motion capture. The training concept representations are obtained via embedding language descriptions of the tasks with T5 (Raffel et al., 2020) and the dummy condition is an embedding of the empty string. During test, we show three to five demonstrations of a composition of training concepts or of a novel concept to a model and ask it to infer the concept. More details are provided in Appendix A.

### 5.1 INFERRING CONCEPTS DESCRIBING SPATIAL **RELATIONS** BETWEEN OBJECTS

Object rearrangement is a common task in robotics (Shah et al., 2018; Yan et al., 2020; Rowe et al., 2019). Here we use a 2D object rearrangement domain to evaluate our method's ability to understand task specification concepts. In this domain, given a concept representing a relation between objects, we generate a single state describing that relation. The concept in a training example describes the relation (either 'right of' or 'above') between only one pair of objects (out of three objects) in the environment. At test time, a model must infer compositions of these pairwise relations and new concepts such as 'diagonal' and 'circle' (see Figure 3a). The results demonstrate that our method can indeed infer unseen compositions and concepts in Table 1. We further show that it can compose new concepts with learned concepts in Table 2 and qualitatively in Figure VII. Note all results are based on inferring two concepts and using a fixed classifier-free guidance weight $\omega = 1.2$.

While successful in most cases, there are also a few failure examples. The accuracy for the 'circle' concept is low (44%). This is most likely due to this concept lying far out of the training distribution. The low accuracy for 'square right of triangle $\wedge$ triangle above circle' (32%) in Table 3 may arise from the combined concept-weight optimization process – as there is no explicit regularization on weights, they may converge to 0 or diverge. However, in Figure VI we show qualitatively that the generated samples are not far from capturing the composed concept.

### 5.2 INFERRING CONCEPTS DESCRIBING **ATTRIBUTES** OF GOAL OBJECTS

We test our method in a goal-reaching agent domain adapted from the AGENT dataset (Shu et al., 2021), where an agent navigates to one of two potential targets in a 3D environment. Conditioned on a concept representing the attributes of the desired target object, and initial state, we generate a state-based trajectory describing an agent navigating to the target. Each object has a color and a shape out of four possible colors and four shapes. There are 16 shape-color combinations for every

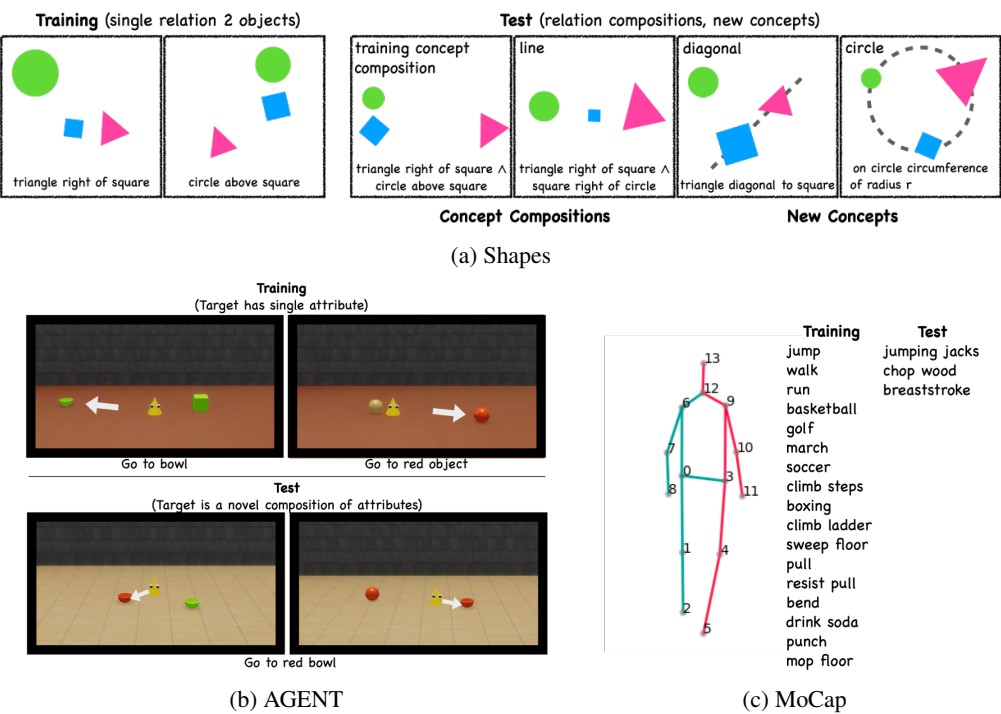

(a) Shapes

(b) AGENT          (c) MoCap

Figure 3: **Experiment Domains.** We demonstrate concept inference for (a) object rearrangement where training demonstrations are single relations between objects ('A right of/above B') and test demonstrations are either compositions of training concepts ('A right of/above B' ∧ 'B right of/above C') or new relations ('A diagonal to B', 'on circle circumference of radius r'). (b) Goal reaching navigation where in training demonstrations, navigation to targets is based on a single attribute (color or shape) and in test demonstrations, navigation is based on a combination of color and shape not present during training. At test time, the distractor object has a combination of attributes within the training distribution. (c) Motion capture where training and test demonstrations describe different sets of human actions such as walking and sweeping vs. swimming.

object, and therefore $16 \times 15/2 = 120$ unique target-distractor combinations. During training, we provide eight such combinations that include all colors and shapes and a concept is conditioned on one of the target's attributes (*e.g.*, color). At test time, we introduce new concepts defined by both target attributes including (1) unseen color-shape target combinations and (2) new target-distractor combinations. Figure 3b shows an example. In training, we have seen targets including a green bowl and a red object. At test time, we encounter a novel composition as a target – red bowl. The distractor objects (green bowl and red sphere) were introduced during training, but they were not paired with a red bowl as the target. As Table 1 shows, our method can successfully generalize to (1) new compositions of target attributes and (2) new target-distractor pairs. It can also generalize to new initial states (Table 2). We report results in these tables for inference of two concepts and fixed classifier-free guidance weight $\omega = 1.6$. Note that the concept 'yellow bowl' in Table 3, where the two concept weights are learned, yields low accuracy (20%), potentially because one of the learned concept weights is close to 0 while the other is 1.1.

## 5.3 INFERRING CONCEPTS DESCRIBING **MOTION**

Unlike prior on Learning to compose human poses from motion capture (MoCap) data (e.g., Wang et al., 2019; Tevet et al., 2023), here we focus on the inverse problem – inferring new actions from MoCap data. In particular, we use the CMU Graphics Lab Motion Capture Database. We train on 17 actions and test inference on three novel concepts (Figure 3c). Learning from demonstrations is especially beneficial in this domain since it could be hard to describe motion concepts in words. We demonstrate qualitatively how our method generates the inferred new concept 'jumping jacks' on new initial states and composes new concepts combining the inferred concept with training concepts 'jump' (Figure 4) and 'march' and 'walk' (Appendix B.2). We also note that there is still space for improvement in the motion generation quality and in the compatibility rate of demonstrations generated by composing inferred and training concepts.

Table 1: **Shapes and AGENT Quantitative Evaluation on Training and Novel Test Concept Inference.** Accuracy of IDM-IGM (ours), goal conditioned BC and VAE over concept generation of training concepts, novel compositions and novel concepts. Full details of our method's evaluation metrics appear in Appendix A and for baseline evaluation in Appendix C. R indicates 'right of' and A indicates 'above'.

| Environment | Setting | Type | BC | VAE | Ours |
|---|---|---|---|---|---|
| | Training | - | **100%** | 60% | **100%** |
| | | $\triangle R\square \wedge \bigcirc A\square$ | 54% | 2% | **94%** |
| | | $\square R\triangle \wedge \bigcirc A\triangle$ | **98%** | 0% | 94% |
| | Composition | $\bigcirc R\square \wedge \triangle A\square$ | **78%** | 30% | 76% |
| Shapes | | $\square R\bigcirc \wedge \triangle A\bigcirc$ | **92%** | 0% | 90% |
| | | line($\bigcirc R\triangle \wedge \triangle R\square$) | 42% | 42% | **100%** |
| | | circle($\bigcirc, \triangle, \square$) | 8% | 0% | **44%** |
| | | diag($\square, \triangle$) | 72% | 0% | **86%** |
| | New Concept | diag($\triangle, \square$) | **94%** | 26% | 90% |
| | | diag($\bigcirc, \triangle$) | 96% | 0% | **98%** |
| | | diag($\triangle, \bigcirc$) | 68% | 18% | **94%** |
| | Training | - | 84% | 89% | **96%** |
| | | red bowl | 60% | **100%** | 80% |
| AGENT | | yellow bowl | **60%** | 55% | 40% |
| | Composition | red cube | 60% | 84% | **100%** |
| | | yellow cube | 60% | **80%** | **80%** |
| | | purple cone | 60% | **87%** | 60% |

Table 2: **Shapes and AGENT Quantitative Evaluation on Diverse Generation of Inferred Novel Concepts.** Accuracy of IDM-IGM (ours), BC and VAE conditioned on inferred novel concepts and new initial states and in composition with training concepts.

| Environment | Setting | Type | BC | VAE | Ours |
|---|---|---|---|---|---|
| | | diag($\square, \triangle$)+$\bigcirc R\square$ | 64% | 0% | **82%** |
| | | diag($\square, \triangle$)+$\bigcirc A\square$ | 12% | 0% | **78%** |
| Shapes | New Concept Composition | diag($\square, \triangle$)+$\bigcirc A\triangle$ | 44% | 0% | **78%** |
| | | diag($\square, \triangle$)+$\bigcirc R\triangle$ | 74% | 0% | **78%** |
| | | diag($\square, \triangle$)+$\triangle A\bigcirc$ | 46% | 0% | **88%** |
| | | red bowl | 52% | **80%** | 72% |
| | | yellow bowl | **50%** | 40% | 48% |
| AGENT | New Initial State | red cube | 54% | 63% | **92%** |
| | | yellow cube | 28% | 60% | **80%** |
| | | purple cone | 48% | 60% | **74%** |

## 5.4 BASELINES

**Goal conditioned behavior cloning.** We compare our method to goal conditioned behavior cloning (BC) which, given a condition and a state, outputs the next state in a sequence. It is trained on our paired pretraining dataset and infers concepts by optimizing the input condition to maximize the likelihood of test demonstration transitions. We test its ability to compose new inferred concepts with training concepts naively by adding conditions that are then inputted into the model. We observe that even though goal conditioned BC has access to the pretrained dataset and conditions, and while it may infer new concepts, it suffers from covariate shift on new initial states, and lacks the ability to generalize to novel compositions (Table 2). In order to achieve these generalization capacities, there is a need for a model that is able to process interpolated (initial states) and composed (concepts) conditions, such as the generative model we use.

**Learning the concept space with a VAE.** We compare our method with a VAE (Kingma & Welling, 2013) that does not utilize the concepts in the paired pretraining data but learns the concept space by reconstructing pretraining data trajectories. The VAE is conditioned on the initial state for AGENT and MoCap domains and concept inference is done by encoding test demonstrations into the learned space to obtain concept $z$. Trajectories are generated via a decoder conditioned on an initial state

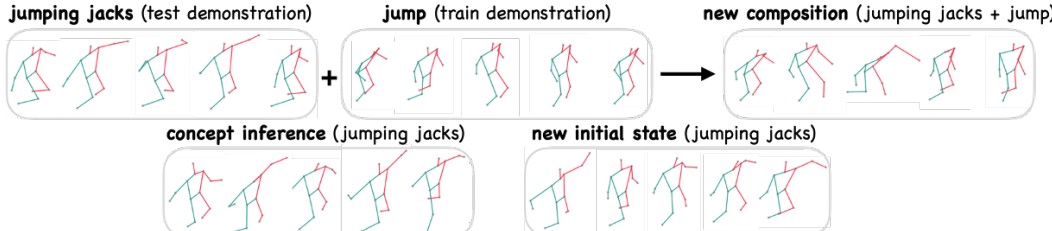

Figure 4: **MoCap Qualitative Visualization.** Generation of 'jumping jacks' conditioned on the newly inferred concept and on a novel initial state, and composed with training concept 'jump'. In the composed generated trajectory, the arms move up and down as in 'jumping jacks' (in 'jump' they are still), and the legs jump forward as in 'jump' (in 'jumping jacks' they move sideways).

and $z$ with added noise. We find that the VAE model learns a latent space that captures training and test concepts in AGENT but does not enable generalization to new initial states. In the shapes domain, the VAE fails to capture the training data as well as novel concept inference. This highlights the importance of concept representations in the pretrained data.

## 5.5 Inferring two concepts yields higher accuracy than one

We repeat the experiments in Tables 1 and 2 for shapes and AGENT and check the effect of the number of learned concepts and their weights. We report results in Table 3, and find that on average, learning two concepts improves concept inference in our experiments. In Figure 5 we show qualitative examples for the MoCap domain on inferring test concepts with one condition. We observe that in 'jumping jacks' the motion lacks the repetitive nature of raising and lowering the arms and appears to transition into the 'walk' action. For 'chop wood' the learned motion lacks the repetitive nature of chopping, and in 'breaststroke' the motion transitions from lying on stomach to standing.

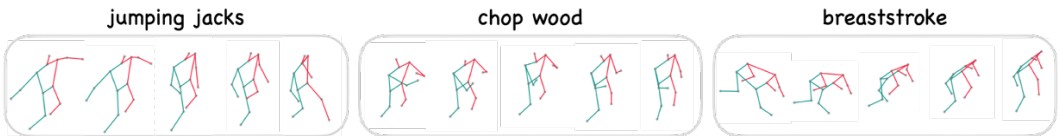

Figure 5: **MoCap Qualitative Ablation.** Generating inferred test concepts with one condition does not capture the demonstrated actions.

## 5.6 What aspects of novel concept compositions are inferred?

We analyze what the learned two concepts in shapes and AGENT infer for novel concept compositions. For each element of a composed concept, (*e.g.*, red and bowl) we generate two sets of 50 samples, each conditioned on a learned concept. In AGENT, generation is also conditioned on initial states sampled based on a composed concept (*e.g.*, red) and the training distribution (in the training data, red targets appear as spheres or cones in combination with yellow sphere or cone distractors). We report accuracy in Table 4 for the generated sets over the composed concepts (42% of the trajectories generated by the first concept target the red object, and 48% of the trajectories generated by the second concept target the red object). In some cases (most notably in the 'line' concept in shapes), each learned condition captures a single composed concept. In other cases, a single learned condition captures both concepts (see Figure VIII). This however is rarely the case, matching our finding in Table 3 that inferring two concepts yields higher accuracy than inferring one concept.

## 6 Discussion

In this work, we formulate the problem of concept inference in decision making as Inverse Decision Making via Inverse Generative Modeling (IDM-IGM). We then develop a novel method for concept inference in decision making based on this new formulation. We evaluate our method against baselines in three domains. Our experimental results show that, unlike the baselines, IDM-IGM can successfully infer novel concepts from a few examples and generalize the inferred concepts to unfamiliar scenarios. It can also compose learned concepts to form unseen decision making pro-

Table 3: **Ablation on Number of Inferred Concepts.** We test the effect of the number of inferred concepts and weights in IDM-IGM on the generation accuracy of concept inference for shapes and AGENT domains. All results are based on learning one and two concepts and their weights. Learning two concepts to represent demonstrations is preferable on average in our experiments.

| Environment | Setting | Type | 1 concept | 2 concepts |
|---|---|---|---|---|
| Shapes | Composition | △R□ ∧ ○A□ | 64% | **76%** |
| | | □R△ ∧ ○A△ | 16% | **90%** |
| | | ○R□ ∧ △A□ | 16% | **88%** |
| | | □R○ ∧ △A○ | **44%** | 32% |
| | | line(○R△ ∧ △R□) | 2% | **90%** |
| | New Concept | circle(○, △, □) | 28% | **42%** |
| | | diag(□, △) | 34% | **80%** |
| | | diag(△, □) | 30% | **88%** |
| | | diag(○, △) | 66% | **88%** |
| | | diag(△, ○) | 88% | **90%** |
| | New Concept Composition | diag(□, △)+○R□ | 36% | **80%** |
| | | diag(□, △)+○A□ | 28% | **68%** |
| | | diag(□, △)+○A△ | 44% | **60%** |
| | | diag(□, △)+○R△ | 52% | **82%** |
| | | diag(□, △)+△A○ | 34% | **78%** |
| AGENT | Composition | red bowl | 60% | **80%** |
| | | yellow bowl | **40%** | 20% |
| | | red cube | 40% | **100%** |
| | | yellow cube | 40% | **80%** |
| | | purple cone | 80% | **60%** |
| | New Initial State | red bowl | **72%** | 66% |
| | | yellow bowl | 54% | **58%** |
| | | red cube | 30% | **86%** |
| | | yellow cube | 50% | **72%** |
| | | purple cone | **68%** | 56% |

Table 4: **Analysis of Learned Concepts in New Composition Inference.** For new composition inference (Table 1) we evaluate what each learned component captures out of demos with two concepts. For both concept 1 and 2, we report the generated accuracy with respect to the concept, when trajectories are generated solely from one learned component. Components are able to successfully represent individual concepts.

| Environment | Concept 1 | | | Concept 2 | | |
|---|---|---|---|---|---|---|
| | Concept 1 | Component 1 | Component 2 | Concept 2 | Component 1 | Component 2 |
| Shapes | △R□ | 42% | **62%** | ○A□ | **88%** | 36% |
| | □R△ | 16% | **64%** | ○A△ | 8% | 8% |
| | ○R□ | 10% | **48%** | △A□ | **86%** | 40% |
| | □R○ | 14% | **90%** | △A○ | 2% | **60%** |
| | ○R△ | 0% | **92%** | △R□ | **64%** | 14% |
| AGENT | red | 42% | **48%** | bowl | **66%** | 56% |
| | yellow | **72%** | 44% | bowl | 58% | **74%** |
| | red | 42% | **76%** | cube | 44% | **96%** |
| | yellow | **40%** | 32% | cube | **82%** | 78% |
| | purple | **48%** | 42% | cone | **56%** | 36% |

cesses thanks to the compositionality of the learned generative models. These results demonstrate the efficacy, sample efficiency, and generalizability of IDM-IGM.

The current evaluation assumes access to ground-truth states. In the future, we plan to investigate the integration of visual perception. Furthermore, we assume that new concepts lie within the landscape of learned concepts in order to infer them from a few demonstrations without retraining the model. We have approached the question of what this landscape is empirically, leaving a theoretical analysis as future work. We are hopeful that with the continued progress in the field of generative AI, more powerful pretrained models become available. Combined with our framework, this will unlock a stronger ability to infer and generalize various decision making concepts in complex domains.

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

## A    DATA GENERATION AND EVALUATION

### A.1    SHAPES

The **training** dataset of $\sim$ 11k samples consists of concepts 'A right of B' and 'A above B', where A and B are one of three objects: circle, triangle or square. Altogether there are 12 possible concepts (two relations and three objects where order is important). In the data generation process, A at center position $(x_A, y_A) \in [0, 5]^2$ with radius $r_A \in [0.3, 1]$ and angle $\theta_A \in [0, 2\pi]$ is considered 'right of' B at center position $(x_B, y_B)$ with radius $r_B$ if $x_A > x_B$ and $|y_A - y_B| \le r_A$. Similarly, A is considered 'above' B if $y_A > y_B$ and $|x_A - x_B| \le r_A$. We further verify that training objects do not overlap. The **test** scenarios include

- Five novel compositions of training concepts (Table 1). 'triangle right of square $\wedge$ circle above square', 'square right of triangle $\wedge$ circle above triangle', 'circle right of square $\wedge$ triangle above square', 'square right of circle $\wedge$ triangle above circle', 'line': 'circle right of triangle $\wedge$ triangle right of square'. There are five demonstrations of each novel composition.

- Five new concepts (Table 1). 'circle': circle, triangle and square lie in equal intervals on the circumference of a circle with radius 1.67 and center $\in [0, 5]^2$, 'square diagonal to triangle', 'triangle diagonal to square', 'circle diagonal to triangle', 'triangle diagonal to circle'. A is considered diagonal to B if their centers lie on $f(x) = x$ and if A is 'above' and 'right of' B. There are five demonstrations of each novel concept.

- Composing a new concept, 'square diagonal to triangle', with five training concepts (Table 2). 'circle right of square', 'circle above square', 'circle above triangle', 'circle right of triangle', 'triangle above circle' weighted by $\omega = 1$ for learned concept weights and $\omega$ for concept inference with fixed weights $\omega$.

We verify that test objects do not overlap and that no training relations unrelated to the specified concepts exist in the demonstrations between objects.

Formally, the **state space** is a 21-tuple describing three shapes (circle, triangle and square) each represented by a 7-tuple: their center 2D position, size, angle and one-hot shape type.

**Evaluation data and metrics**. For training, we generate 50 states conditioned on uniformly sampled training concept embeddings for single relations. For each concept composition and new concept, we report accuracy on 50 generated states conditioned on the learned concept. For new concepts composed with training concepts we report accuracy based on 50 generated states for each training concept. Results in Tables 1 and 2 are reported for best $\omega \in \{1.2, 1.4, 1.6, 1.8\}$. We report accuracy based on a relaxed version of the data generation process: A is considered 'right of' B if $x_A > x_B$ and $|y_A - y_B| \le 2 \cdot \max\{r_A, r_B\}$, A is considered 'above' B if $y_A > y_B$ and $|x_A - x_B| \le 2 \cdot \max\{r_A, r_B\}$. A, B and C lie on a 'circle' if their centers form a circle of radius $r$ where $|r - 1.67| < 0.3$ and A is 'diagonal' to B if $x_A > x_B$, $y_A > y_B$ and they lie within a $2 \cdot \max\{r_A, r_B\}$ margin of $f(x) = x$.

### A.2    AGENT

To collect training and test demonstrations, we follow the data generation process in the AGENT benchmark environment (Shu et al., 2021) which provides a planner for navigation given the desired target. In the provided environment, the agent's initial position $a_{t=0}^p = (0, 0.102, -3.806)$, color (yellow) and shape (cone) are fixed. Each target has a color $t_1^c, t_2^c \in \{\text{red}, \text{yellow}, \text{purple}, \text{green}\}$, a shape $t_1^s, t_2^s \in \{\text{cone}, \text{sphere}, \text{bowl}, \text{cube}\}$ and a position $t_1^p \in [0, 1.66] \times \{0.102\} \times [-4.355, -3.257]$, $t_2^p \in [-1.66, 0] \times \{0.102\} \times [-4.355, -3.257]$. The **training** dataset of $\sim$ 900 samples consists of concepts 'go to red object' and 'go to yellow object' where $t_1^c \in \{\text{red}, \text{yellow}\}$, $t_2^c \in \{\text{red}, \text{yellow}\} \setminus \{t_1^c\}$ and $t_1^s, t_2^s \in \{\text{cone}, \text{sphere}\}$, and concepts 'go to bowl' and 'go to cube' where $t_1^s \in \{\text{bowl}, \text{cube}\}$, $t_2^s \in \{\text{bowl}, \text{cube}\} \setminus \{t_1^c\}$ and $t_1^c, t_2^c \in \{\text{purple}, \text{green}\}$. The **test** scenarios include

- Five novel compositions of training target color and shape attributes (Table 1). 'go to red bowl' $(t_i^c, t_j^c = \text{red}, t_i^s = \text{bowl}, t_j^s \in \{\text{cone}, \text{sphere}\}$ or $t_i^s, t_j^s = \text{bowl}, t_i^c = \text{red}, t_j^c \in \{\text{purple}, \text{green}\}$ where $i \in \{1, 2\}$ and $j = \{1, 2\} \setminus i$), 'go to yellow bowl'

$(t_i^c, t_j^c = \text{yellow}, t_i^s = \text{bowl}, t_j^s \in \{\text{cone}, \text{sphere}\}$ or $t_i^s, t_j^s = \text{bowl}, t_i^c = \text{yellow}, t_j^c \in \{\text{purple}, \text{green}\})$, 'go to red cube' $(t_i^c, t_j^c = \text{red}, t_i^s = \text{cube}, t_j^s \in \{\text{cone}, \text{sphere}\}$ or $t_i^s, t_j^s = \text{cube}, t_i^c = \text{red}, t_j^c \in \{\text{purple}, \text{green}\})$, 'go to yellow cube' $(t_i^c, t_j^c = \text{yellow}, t_i^s = \text{cube}, t_j^s \in \{\text{cone}, \text{sphere}\}$ or $t_i^s, t_j^s = \text{cube}, t_i^c = \text{yellow}, t_j^c \in \{\text{purple}, \text{green}\})$, 'go to purple cone' $(t_i^c, t_j^c = \text{purple}, t_i^s = \text{cone}, t_j^s \in \{\text{bowl}, \text{cube}\}$ or $t_i^s, t_j^s = \text{cone}, t_i^c = \text{purple}, t_j^c \in \{\text{red}, \text{yellow}\})$. Note that in each scenario the distractor object either has the same color or shape as the target, and combined with its other attribute (shape or color), it is within the training distribution (*i.e.*, red and yellow cones and spheres, and purple and green bowls and cubes). There are five demonstrations of each novel composition.

- Conditioning each novel composition concept on novel initial states sampled from the test distribution (Table 2).

The **initial state space** we condition on is a 52-tuple based on the state space in Shu et al. (2021) describing the agent and two objects. The agent is represented by its 3D position, quaternion $\in \mathbb{R}^4$, velocity $\in \mathbb{R}^3$, angular velocity $\in \mathbb{R}^3$, one-hot type (representing the agent and two objects), rgba color, and one-hot shape (representing the four possible shapes). Similarly, each object is represented by its 3D position, type, color and shape.

The **demonstration state space** is a 13-tuple representing the first 13 dimensions of the initial state space (agent position, quaternion, velocity and angular velocity). Demonstrations $i \in [N]$ have horizons $H_i \leq 150$ and are padded to length 150 using the final state. During training, we sample the demonstrations to generate subtrajectories of length 128.

**Evaluation data and metrics**. For training, we generate 50 trajectories conditioned on uniformly sampled training concepts and initial states for single attributes. For each concept composition, since the initial state determines the optimal plan deterministically, we generate trajectories conditioned on the inferred concept and demonstrated five test initial states for each concept. For evaluating each concept composition on new initial states, we generate trajectories conditioned on the learned concept and 50 initial states sampled from the test distribution. Results in Tables 1 and 2 are reported for best $\omega \in \{1.2, 1.4, 1.6, 1.8\}$. We report accuracy based on whether the agent in the generated trajectory has made progress towards the desired target ($|a_{t=128}^p - t^p| < |a_{t=0}^p - t^p|$).

## A.3 MOCAP

We select 17 training motion categories and three test categories from the 144 subjects in the dataset. We discard videos with less than 256 frames and those which contain more than one human action per trajectory.

The **training** set contains $\sim 250$ demonstrations of the following human actions: 'jump' (72 demonstrations), 'walk' (69), 'run' (16), 'basketball' (13), 'golf' (11), 'march' (11), 'soccer' (9), 'climb steps' (9), 'boxing' (8), 'climb ladder' (6), 'sweep floor' (5), 'pull' (4), 'resist pull' (4), 'bend' (3), 'drink soda' (3), 'punch' (2) and 'mop floor' (2). The **test** scenarios include

- Three new concepts. 'jumping jacks' (3), 'chop wood' (2), and 'breaststroke' (3).

- Conditioning each novel composition concept on novel initial states sampled from novel subjects performing the test action ('jumping jacks' and 'breaststroke'), and for novel states sampled from the demonstration test distribution ('chop wood').

- Composing new concept 'jumping jacks' weighted by learned concept weights with three training concepts ('jump', 'march' and 'walk') weighted by $\omega = 1$, and conditioned on a training initial state from the training concepts (Figures 4, and IX).

The **initial state space** is a 42-tuple representing the 3D position for 14 joints (see Figure 3c). The original dataset contains 31 joints, our version is adapted with Tanke et al. (2021) to reduce the number of joints to 14 and to remove rotation and translation.

The **demonstration state space** is the same with horizons $H_i$, $i \in [N]$. We subsample the original trajectories every 8 steps after which the training dataset has demonstrations of length $\in [33, 1569]$ with mean length and standard deviation $136 \pm 213$. The test demonstrations have the following length range, mean and standard deviation: jumping jacks are all of length 64, chop wood

$\in [74, 136]$, $105 \pm 31$, and breaststroke $\in [195, 323]$, $242 \pm 57$. We further sample trajectories to generate subtrajectories of length 32 which we train on.

**Evaluation data and metrics**. For each new concept, we generate one trajectory conditioned on the learned concept and an initial demonstrated test state. For evaluating each new concept on new initial states, we generate trajectories conditioned on the learned concept and a novel initial test state sampled from novel subjects in the CMU Graphics Lab Motion Capture Database ('jumping jacks' and 'breaststroke') or from the original test trajectories subsequences that were discarded during the subsampling data processing ('chop wood'). For new concepts composed with training concepts, we generate trajectories each conditioned on the learned concept, a uniformly sampled training concept and training initial state. We present some successful qualitative results in Figures 4, and IX.

# B    ADDITIONAL RESULTS

## B.1    QUALITATIVE EVALUATION OF SHAPES

In Figure VI we show generated states for new composed concept inference. In Figure VII we show generated states for a novel concept, and its composition with a training concept. In Figure VIII we show generated states for single inferred components from concept compositions.

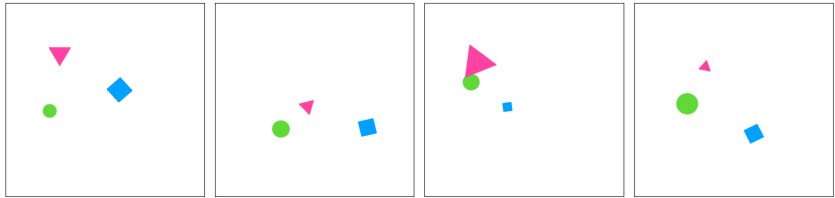

Figure VI: **Shapes New Composition Qualitative Evaluation.** 'square right of circle $\wedge$ triangle above circle' has low accuracy in Table 1, yet in practice generated states with accuracy 0% (second and last from left) are not far from capturing the concept.

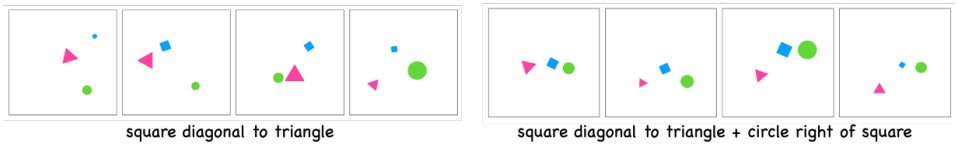

Figure VII: **Shapes New Concept Qualitative Evaluation.** Inferring new concept 'square diagonal to triangle' and composing it with training concept 'circle right of square'.

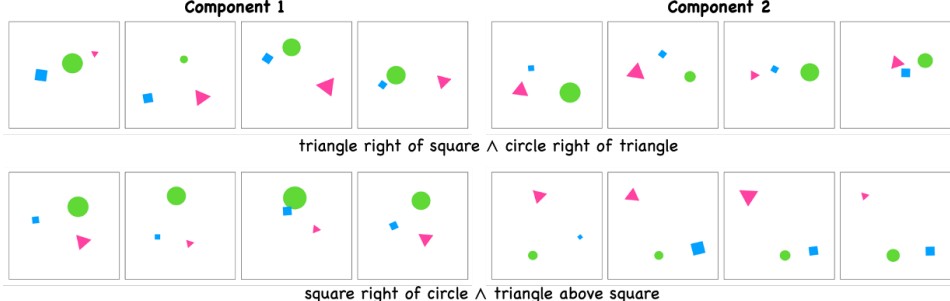

Figure VIII: **Shapes New Compositions Analysis Qualitative Evaluation.** 'triangle right of square $\wedge$ circle right of triangle' each learned component corresponds to a single composed concept. In 'square right of circle $\wedge$ triangle above square', learned component 2 corresponds to both composed concepts and component 1 to none.

### B.2 QUALITATIVE EVALUATION OF MOCAP COMPOSITIONS

In Figure IX we show more results on composing inferred concept 'jumping jacks' with training concepts 'march' and 'walk'.

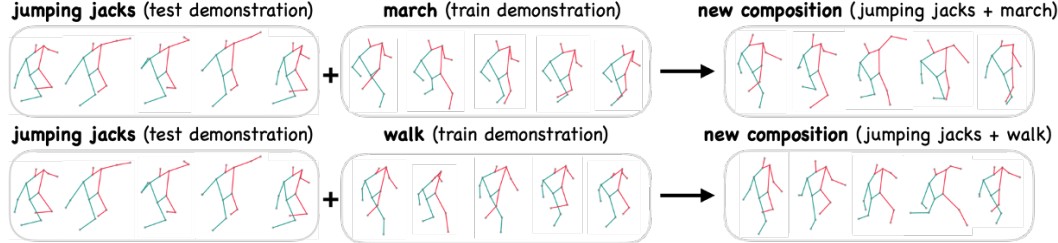

Figure IX: **MoCap Qualitative Visualization.** We compose the novel inferred concept in MoCap ('jumping jacks') with training concepts ('march' and 'walk'). In the composed trajectories, the arms move up and down as in 'jumping jacks' (in 'march' and 'walk' they are still), and the legs march forward as in 'march' and walk forward as in 'walk' (in 'jumping jacks' they move sideways).

## C  IMPLEMENTATION DETAILS

**Diffusion model.** We represent the noise model $\epsilon_\theta$ with an MLP for the object rearrangement domain and with a temporal U-Net for the goal-reaching agent and motion capture domains as implemented in Ajay et al. (2023). We use the same hyperparameters as in Ajay et al. (2023) with the exception of the probability of removing conditioning information, $p$, which we set to $0.1$.

**Baselines.** We implemented behavior cloning (BC) (Pomerleau, 1989) and a conditional variational autoencoder model (CVAE) (Kingma & Welling, 2013). Both models use MLPs for the architecture and we used AdamW (Loshchilov & Hutter, 2017) for optimizing them with a learning rate of $6 \cdot 10^{-4}$. The BC model is deterministic, $s_{t+1} = \pi(s_t, c_t)$. In the shapes dataset, we only condition on $c_t$. In order to add stochasticity, when evaluating BC on inferring conditions and generating states, we average results over 50 different seeds. For our CVAE experiments, we input the initial state to the encoder and decoder of the VAE. However, when evaluating our the CVAE on the test dataset, we are not able to use the conditions from the dataset because the learned latent space of the CVAE is guaranteed to be compatible with the latent space from the dataset. Consequently, for the text description of the task, we find the most similar state or trajectory from the training set, and encode it to obtain a latent. From there we concatenate that latent with the new initialization. We sample 50 trajectories per initialization and average the results. For the compositional experiments we simply add the the learned conditions with the given conditions.

