# OpenReview forum: "Inverse Decision Making via Inverse Generative Modeling"
_ICLR.cc/2024/Conference — ICLR 2024 Conference Withdrawn Submission_

### Official Review · Reviewer_veLU · 2023-10-20

**Soundness:** 2 fair
**Presentation:** 2 fair
**Contribution:** 2 fair
**Rating:** 3
**Confidence:** 3

**Summary:**

This paper considers learning an inverse generative model for behavior datasets. This allows inferring a latent "concept" explaining a demonstrated trajectory. The inferred concept can then be combined with new initial states to generate new trajectories exhibiting the same concept. The latent space is learned such that different behaviors demonstrated by different trajectories can be combined and the combined trajectory generated. The learned model is tested in open loop trajectory generation tasks.

**Strengths:**

## Contribution
- As the authors note, inverse learning in decision-making problems is a difficult problem where the current methods have significant drawbacks. Applying inverse generative models seems like good idea and worthy of investigation.
- The learned concepts do seem to capture regularities in the demonstrated behaviors.

**Weaknesses:**

## Presentation
- In the introduction, there is a claim, that concept inference in decision-making requires understanding of an agent's internal goals. This may be true in general, but I doubt that it applies to the problems considered in this paper. That's because given the concept, the trajectories appear to be more or less Markovian. It would be good to discuss this claim with respect to the contributions of the paper.

## Novelty
- The proposed method seems very similar to the one proposed in [1]. Considering that this paper focuses on open-loop trajectory generation, I don't see how this is fundamentally different from generating images or any other kind of non-interactive data. The fact that the training data was generated by some process that has temporal structure informs the architecture selection here, but it doesn't change the generation problem on an abstract level compared to say generating images.

## Soundness
- The "Formulation" section lacks in detail. The "concept" is never defined formally.
- The formulation compares the proposed method to inverse RL. However, it targets an entirely different problem setting than IRL, since it does not consider closed-loop control. Generating open-loop trajectories circumvents the biggest challenge in learning from demonstrations, because if the learned policy is never deployed in an environment, it will not face the distribution shift that makes learning from demonstrations hard in the first place. The related work [2] shows that you can learn an action-generating policy by using the data generated by the diffusion model, so we know it is possible. However, there's no demonstration in this paper that the learned generative model can be used for learning policies that work in the environments where the data was generated.

**Questions:**

- How is the proposed concept discovery algorithm different from prior work? Specifically, equation (2) looks very similar to equation (11) in [1].

[1] Liu et al., 2023., Unsupervised Compositional Concepts Discovery with Text-to-Image Generative Models

[2] Ajay et al., 2023, Is Conditional Generative Modeling All You Need For Decision-Making?

---

### Official Review · Reviewer_SoYM · 2023-11-01

**Soundness:** 3 good
**Presentation:** 3 good
**Contribution:** 2 fair
**Rating:** 3
**Confidence:** 3

**Summary:**

This paper primarily presents an inverse decision reasoning approach based on generative modeling for inferring concepts in the decision-making process. An diffusion model is first trained on a large pretraining dataset and then the diffusion model is frozen to infer the new concept given an new demonstration dataset. The proposed method shows promising results on three domains: object rearrangement, goal-oriented navigation, and motion capture.

**Strengths:**

1. The paper demonstrates the potential of diffusion model in the context of sequence inference, expanding the boundaries of diffusion capabilities.
2. The writing and presentation of the paper are exceptionally clear.

**Weaknesses:**

1. The setup of the paper involves training on one dataset $D_{pretrain}$ and then reasoning on a demonstration dataset $D$. While this paradigm is different from IL or IRL, it bears strong similarities to traditional supervised learning.
2. The method lacks novelty and seems to directly apply the diffusion model to this problem, which appears relatively straightforward without significant challenges.

**Questions:**

1. Compared to behavioral cloning, the method proposed in the paper does show some improvements. However, has this improvement come at the cost of increased computational complexity and inference time? This is an important aspect that should be considered, especially given the time-consuming nature of diffusion model inference. The paper should also consider the inference time of different baselines.
2. Additionally, it should clearly delineate the differences from Decision Diffuser, particularly in experiments  conditioned on skills and constraints.

---

### Official Review · Reviewer_Mjji · 2023-11-03

**Soundness:** 3 good
**Presentation:** 3 good
**Contribution:** 2 fair
**Rating:** 6
**Confidence:** 4

**Summary:**

In this work, the authors formulate the problem of concept inference as Inverse Decision Making via Inverse Generative Modeling (IDM-IGM). Specifically, they develop a novel method based on conditional diffusion models for concept inference in decision-making problems (such as attributes of target objects, motions, etc.). Their method is evaluated against goal-conditioned BC and VAE in two static datasets and one human motion dataset. They show that their approach can infer novel concepts from a few demonstrations at test time and they can generate new demonstrations based on the inferred concepts.

**Strengths:**

+ An intriguing formulation of concept inference with diffusion models
+ The method is well-motivated and reasonable
+ The results of inferring novel concepts and then generating agent motion in unseen environments seem promising
+ The problem the authors try to solve can have a great impact on a broader community (e.g., robotics, embodied AI).

**Weaknesses:**

Overall I like the direction and the idea of this paper. However, the experiment evidence can be stronger by including more non-static dataset results (like the human motion one). The Shapes dataset is more like a 2D toy dataset. I believe a potential usage of this approach is for producing novel trajectories for AI agents in general (cars in the domain of AV, mobile manipulators in the domain of robotics, etc.). It would be much stronger if the authors could show (even some preliminary) results on domains like these.

**Questions:**

See “weaknesses”

---

### Official Review · Reviewer_oDMW · 2023-11-05

**Soundness:** 2 fair
**Presentation:** 2 fair
**Contribution:** 2 fair
**Rating:** 3
**Confidence:** 2

**Summary:**

This paper proposes a problem formulation where the goal is to infer latent concepts from state trajectories. More precisely, the learner is given a trajectory $\tau = s_1, ..., s_T$ conditioned on some latent concept $c$ (for example, a goal or behavior specification), and the goal is to infer $c$ from $\tau$. They propose to solve this problem by pretraining a generative model on a large dataset of $(c, \tau)$ pairs, and then at test time, given a trajectory $\tau_{test}$, optimizing the corresponding $c_{test}$ by backprop through the generative model. They find that by pretraining on a large dataset, the model is able to learn good interpolation behavior that enables it to compositionally generalize to novel concepts at test time. They evaluate this approach on a toy shape dataset, the AGENT dataset and a MoCap dataset.

This paper has some nice ideas, but the relationship with sequential decision-making is unclear despite being one of the important motivations in the intro. Also, there are no comparisons to in-context methods for sequential decision-making, which address similar problems. Therefore, I don't think the paper is ready for publication in its present form.

**Strengths:**

- The problem that this paper addresses, namely inferring latent concepts from a small number of demonstrations, is relevant and interesting. I think their approach of leveraging large pretraining datasets is reasonable, and not too much of a limiting assumption.
- To my knowledge, the approach is novel, although I am less familiar with the generative modeling literature, so I am not sure of this.

**Weaknesses:**

- The relationship to sequential decision making is not very clear. It seems like the problem generally deals with sequences, and it's not clear where the sequential decision-making aspect comes in. Although the first part of the paper motivates the problem through the sequential decision-making setting, the experiments for the most part deal with generalization of object properties and relations, and it's not very clear where the sequential decision-making comes in. It would be helpful to clarify what the state and action spaces are here, since this is the setting introduced in Section 3. The performance metric in all the tables is accuracy of the decoded concepts - how does this relate to the policy performance?

- There are no comparisons to in-context approaches for offline RL, which seems to be closely related to the problem this paper is addressing. Two other methods which the proposed approach should be compared to are [1] and those discussed in [2]. Both of these use large pretrained datasets to enable new behaviors at test time in a few-shot manner.

[1] https://arxiv.org/abs/2206.13499
[2] https://openreview.net/forum?id=MYEap_OcQI

**Questions:**

What are the state and action spaces in each of the experiments? This is not clear. How is the VAE baseline used to modify the policy network? Is the concept vector inferred by the proposed vector used to condition the policy at test time? This is not clear.